# HOX Gene Expressions in Cultured Articular and Nasal Equine Chondrocytes

**DOI:** 10.3390/ani11092542

**Published:** 2021-08-30

**Authors:** Christiane Storch, Herbert Fuhrmann, Axel Schoeniger

**Affiliations:** Institute of Biochemistry, Faculty of Veterinary Medicine, University of Leipzig, An den Tierkliniken 1, 04103 Leipzig, Germany; christiane.storch@uni-leipzig.de (C.S.); fuhrmann@vetmed.uni-leipzig.de (H.F.)

**Keywords:** HOX genes, equine, cartilage, chondrocytes, nasal, articluar, culture

## Abstract

**Simple Summary:**

Once articular cartilage is damaged, it is unable to regain its original tissue integrity, which leads to osteoarthritis including degeneration of the joint, suffering and pain. In equine medicine there is no therapy available to repair joint defects. Hyaline cartilage of nasal septum shows a high basal collagen II expression, which may have a positive effect on damaged articular cartilage. Therefore, nasal septum could be a potential source for chondrocytes for autologous implantation in the future.

**Abstract:**

Osteoarthritis the quality and span of life in horses. Previous studies focused on nasal cartilage as a possible source for autologous chondrocyte implantation (ACI) in cartilage defects in humans. “HOX gene-negative” nasal chondrocytes adapted articular HOX patterns after implantation into caprine joint defects and produced cartilage matrix proteins. We compared the HOX gene profile of equine chondrocytes of nasal septum, anterior and posterior fetlock to identify nasal cartilage as a potential source for ACI in horses. Cartilage was harvested from seven horses after death and derived chondrocytes were cultured in a monolayer to fourth subcultivation. HOX A3, D1, D8 and chondrocyte markers COL2 and SOX9 were analyzed with qPCR in chondrocytes of three different locations obtained during passage 0 and passage 2. HOX gene expression showed no significant differences between the locations but varied significantly between the horses. HOX genes and SOX9 remained stable during culturing. Cultured nasal chondrocytes may be a target for future research in cell-based regenerative therapies in equine osteoarthritis. The involvement of HOX genes in the high regenerative and adaptive potential of nasal chondrocytes observed in previous studies could not be confirmed.

## 1. Introduction

Diseases affecting articular cartilage still pose a challenge to clinicians and researchers today. Cartilage is a tissue with a low metabolic rate, limited regeneration capacity and high susceptibility to degenerative processes due to trauma and aging [1,2], leading to osteoarthritis in horses and humans [3,4]. Osteoarthritis in horses often limits the life expectancy [5] and leads to high medical costs and economical losses [6]. As these animals are used for sport or leisure activities, common articular therapies in equine medicine such as microfracture are unsatisfying because their histological outcomes resemble those of untreated lesions. [7]. Horses are also considered an animal model for translational research because their cartilage composition is similar to that of humans, and equine joints are exposed to tremendous load and stress [3].

Promising cell-based therapies have been explored in recent years to restore chondrocyte function, many of them autologous-cell-based [8]. Mesenchymal stem cells (MSCs) have been studied intensively, as these cells are easy to isolate from various tissues [9] and demonstrated chondrogenic differentiation in culture [10]. In vivo studies revealed that MSCs become hypertrophic and calcified [11] and form fibrocartilage-like tissue after implantation [12]. Since fibrous cartilage is vulnerable to mechanical joint forces in long-term studies [1] and thus are prone to complications such as chip formation or re-tearing at the implantation site, MSC-based therapies have not fulfilled the requirements for clinical approval.

In contrast, autologous chondrocyte implantation (ACI)-based therapy is already FDA approved [13]. Autologous chondrocytes form hyaline-like cartilage [14] and demonstrated satisfying long-term MRI and clinical outcomes [15]. Nevertheless, ACI-based therapies remain insufficient as the gold standard, since donor cartilage is required from a less affected joint [16].

In comparison with ACs, nasal chondrocytes (NCs) are easy to obtain and cause minimal morbidity at the donor site [17]. NCs originate from the neural-crest in contrast to mesoderm-derived ACs [8]. The identification and differentiation of tissues are largely determined by specific homeobox (HOX) gene profiles [18]. Pelttari et al. (2014) investigated HOX gene profiles in human and caprine nasal and articular cartilage. They found that only a few HOX genes were active in nasal cartilage (HOX negative), while many HOX genes were activated in the joints. Pelttari et al. (2014) treated artificial articular joint defects of goats with autologous NCs that were expanded in a chondrogenic culture before implantation. Three and six months after implantation, the NCs produced hyaline matrix proteins and matched the articular HOX gene profile. The group concluded that the negative HOX gene profile of nasal cartilage may be crucial for their adaptability [19]. Accordingly, chondrocytes retrieved from nasal cartilage present a potential therapy for articular cartilage defects [20]. Since nasal cartilage has already been used for the reconstruction of nostrils in humans, safety concerns are low [21].

Despite the high degree of evolutionary conservation of HOX genes [22], HOX D8 of nasal chondrocytes exhibited species-specific expression in humans and goats [19]. Furthermore, NCs produce a functional cartilagous matrix [23] and seem to be more robust than MSCs and ACs with respect to the effect of inflammatory factors [24].

We determined the HOX gene profile of equine chondrocytes obtained from the nasal septum, anterior and posterior fetlock joints during cultivation in a monolayer. Additionally, we measured the gene expression of the chondrocyte markers SOX9 and COL2 [25,26]. The aim of this study was to investigate cultured nasal chondrocytes as a potential therapeutic option for equine osteoarthritis by analyzing the HOX gene profile in these cells. To our knowledge, no study has been carried out on HOX genes in equine cartilage so far.

## 2. Materials and Methods

### 2.1. Animals

Cartilage samples were collected from seven horses (three females, two males, two neutered males). Ages ranged from 1 to 23 years, whereby in one case, age was estimated to at least 15 years based on dentition. Six horses were euthanized for medical reasons, and one died of tetanus at the Department for Horses, Faculty of Veterinary Medicine, Leipzig University. Subsequently, the animals were transported to the Institute of Pathology, Faculty of Veterinary Medicine, Leipzig University, and were stored in a cooling chamber until sample collection.

### 2.2. Isolation of Cartilage

Articular cartilage and the nasal septum were isolated in the Department for Veterinary Pathology of the University of Leipzig. The collection of cartilages from each individual started between 6.25 and 19.5 h after death from one front fetlock, one hind fetlock and the nasal septum. Separate dissection kits were used for each layer. After the opening, the joints were checked for integrity of the cartilage. Only two horses showed superficial erosions in both joints. No cartilage was isolated from these defects. Immediately after the retrieval of a layer, the remaining tissues were rinsed several times with 1X PBS and gloves were changed. Cartilage was exfoliated in pieces of 3 to 4 mm with a scalpel, rinsed several times with 1X PBS and transferred into a sterile tube with transport solution containing 1X PBS, 1% penicillin/streptomycin (Biochrom, Berlin, Germany) and 0.8% amphotericin B (Sigma-Aldrich^®^, St. Louis, MO, USA).

### 2.3. Cell Dissociation

Enzyme solutions containing 1X PBS with 1.0 mg/mL collagenase CLS II (Biochrom, Berlin, Germany, sterile-filtered), 0.5 mg/mL collagenase P (Sigma-Aldrich^®^, St. Louis, MO, USA, sterile-filtered), 0.1 mg/mL hyaluronidase (Sigma-Aldrich^®^, St. Louis, MO, USA, sterile-filtered), 10% heat-inactivated fetal bovine serum (FBS, Biochrom, Berlin, Germany), 1% penicillin/streptomycin (Biochrom, Berlin, Germany) and 0.9% amphotericin (Sigma-Aldrich^®^, St. Louis, MO, USA) were mixed directly before isolation of the cartilage and kept at 4 °C until usage. Transport solutions were discarded carefully before adding the enzymatic solution to each tube with the cartilage slices. The samples were incubated at 37 °C and shaken at 108 rpm (Heidolph^®^ unimax 1010, Incubator 1000, Schwabach, Germany) for 18 to 19 h. After digestion, the enzymatic solution was diluted with 1X PBS (37 °C), filtered through a 100 µm cell strainer and washed twice with 1X PBS at room temperature (Multifuge^®^ 3 S-R, Heraeus^®^, Hanau, Germany, at 1200 rpm for 6 min) before suspending the chondrocytes in 2 mL of culture medium (37 °C).

### 2.4. Cell Culture

Chondrocytes were counted microscopically (Zeiss^®^ Axiovert^®^, Oberkochen, Germany, 100×) with an improved NEUBAUER chamber with Trypan Blue (Sigma-Aldrich^®^, St. Louis, MO, USA). Cells were seeded in 25 cm${}^{3}$ cell culture flasks coated with 1 mL of FBS (37 °C) at a density of 5000 cells per cm${}^{2}$ and cultured in 4 mL of Dulbecco’s Modified Eagle Medium (DMEM) with 4500 mg/L glucose, L-glutamine, sodium bicarbonate, without sodium pyruvate (Biochrom, Berlin, Germany) supplemented with 50.0 mg/L phospho-2-ascorbate (Sigma-Aldrich^®^, St. Louis, MO, USA), 0.9 mL/L human insulin (Sigma-Aldrich^®^, St. Louis, MO, USA) and 100 µL/L of a 20-fold diluted 100× non-essential amino acids stock solution (Sigma-Aldrich^®^, St. Louis, MO, USA) with 10% heat inactivated FBS (Biochrom, Berlin, Germany), 1% penicillin/streptomycin (Biochrom, Berlin, Germany) and 0.8% amphotericin B (Sigma-Aldrich^®^, St. Louis, MO, USA) at 37 °C in 10% CO_2_ and relative air humidity of 95%. Cultures were checked by macroscopic and microscopic inspection (Zeiss^®^ Axiovert^®^, Oberkochen, Germany, 25×, 100× and 400×) every 3 or 4 days. Chondrocytes were fed with 2 mL of fresh medium, or complete medium (37 °C) was exchanged depending on the adherence of cells.

### 2.5. Cell Passage

All cultures of each horse were subcultivated on the same day to ensure comparability between all three locations in one individual. Cells were subcultivated when all three cultures reached a minimum confluence of 50%. An amount of 0.05% Trypsin-EDTA was added to the cultures and the flasks were incubated for 6 min to detach chondrocytes. Subsequently, the flasks were shaken (KS-15 control, Edmund Bühler^®^, Bodelshausen, Germany) at room temperature for 1 min to increase the cell yield. Following two washing steps with 1X PBS followed at room temperature, cells were counted, seeded and cultured as described above, or cryoconserved at −80 °C with 10% DMSO (AppliChem^®^, Darmstadt, Germany) for further use.

### 2.6. RNA Extraction

RNA extraction was performed by using the Monarch Total RNA Miniprep Kit (New England Biolabs^®^, Ipswich, MA, USA). Solutions and the samples were stored on cooling blocks. Chondrocytes were obtained from passage 0, designated as T1, and from passage 2, designated as T2. A total of 300.000 to 400.000 cells were transferred in 2 mL of RNase-free Eppendorf^®^ tubes, washed two times with 1X PBS and centrifuged at 500× *g* (Eppendorf^®^ Centrifuge 5417R, Hamburg, Germany). RNA was extracted following the manufacturer’s instructions for cultured mammalian cells, including the recommended DNase I treatment. A two-step elution of RNA was performed, whereby 18 µL of RNase-free water was added to the spin columns, followed by centrifugation. This was repeated to maximize the RNA yield and concentration. The columns were spun for 30 s during the first step and for 1 min during the second step. RNA solutions were aliquoted and stored at −80 °C for further use.

### 2.7. RNA Concentration and Quality

All samples were stored on dry ice. RIN was measured with a Fragment Analyzer at the Interdisciplinary Centre for Clinical Research of Leipzig University. Of the 42 samples, 11 were selected randomly for RIN measurement and showed high quality (mean 8.6, SD = 1.19). The RNA concentrations and A260/A280 ratios of all samples were measured with a NanoPhotometer^®^ (Implen^®^, Munich, Germany) with a lid-factor of 10 at the Division of Molecular Biological–biochemical Processing Technology of the Centre for Biotechnology and Biomedicine Centre of Leipzig University. The mean A260/A280 ratio was 1.98 (SD = 0.07).

### 2.8. cDNA Synthesis

cDNA synthesis was carried out with the Protoscript II First Strand cDNA Synthesis kit (New England Biolabs^®^, Ipswich, MA, USA) following the manufacturer’s recommendations using Oligo-d(T)-primers. A total of 375 ng of total RNA was transcribed into cDNA. For no-reverse transcription (no-RT) controls, 150 ng of total RNA was used. Incubations were performed with an Eppendorf Thermomixer comfort. cDNA samples were aliquoted and stored at −20 °C for further use.

### 2.9. Primer Design and Testing

Species-specific primers spanning across exon–exon boundaries were designed based on nucleotide sequences available in the NCBI database using Primer-BLAST (https://www.ncbi.nlm.nih.gov/tools/primer-blast/, accessed on 18 December 2018) and are listed in Table 1. The selection of the reference genes was based on a previous publication [27]. According to the NCBI database, all gene sequences were predicted, except for GAPDH and COL2. Primers were synthesized and HPLC purified by Metabion^®^ (Planegg, Germany). In accordance with the MIQE Guidelines [28], primers were tested, including a qualitative PCR which was carried out with a T-Gradient ThermoBlock (Biometra^®^, Goettingen, Germany) using FastGene^®^ Optima HotStart Ready Mix (Nippon Genetics^®^, Dueren, Germany) and the following settings: 95 °C for 3 min (activation of Taq polymerase), followed by 50 cycles of denaturation at 95 °C for 30 s, annealing at 60 °C for 30 s, elongation at 72 °C for 30 s and a final inactivation step at 72 °C for 3 min. Amplicons were analyzed by electrophoresis in 1% TAE buffer at 60 V for 50 min on a 1.8% agarose gel containing GelRed^®^ (Biotium, Fremont, CA, USA). Documentation was carried out using a GeneGenius Bio Imaging System (Syngene^®^, Cambridge, UK), GeneSnap (Syngene^®^, Cambridge, UK, version 7.12.06) and GeneTools software (Syngene^®^, Cambridge, UK, version 4.03.05.0). Primer efficiency tests were carried out under experimental conditions. Efficiencies of 0.79 to 1.12 were accepted. After the primer evaluation, five genes of interest (GOI), including three HOX genes as well as two chondrocyte marker genes, and two reference genes were retained in the study.

### 2.10. qPCR

qPCR was run in a Rotor-Gene Q^®^ (Qiagen^®^, Hilden, Germany) using a SensiFAST SYBR^®^ No-Rox Kit (Bioline, Meridian Bioscience^®^, London, UK) containing SYBE Green I, 3 nM Mg_2_Cl and antibody-mediated hot-start DNA polymerase following the manufacturer’s instructions. The optimal amount of template was 200 ng. The best primer performance was observed at a concentration of 100 nM for GAPDH, RPL32 and COL2 and 200 nM for HOX A3, D1, D8 and SOX9. NTCs were included in every run. No-RTs were carried out in a separate run with one positive control and NTCs. The reaction mix was heated to 95 °C for 2 min to activate polymerase. This was followed by 40 cycles at the following settings: 95 °C for 15 s (melting), 55 °C for 30 s (annealing), 72 °C for 15 s (elongation). A melting curve was recorded between 55 and 99 °C. Three technical replicates were prepared for each gene of each horse.

### 2.11. Data Analysis

Data were analyzed with the Q-Rex software (Qiagen ^®^, Hilden, Germany, 2013). The threshold was set to 0.05 for the CT values and melting curves were checked for unspecific products and primer dimers. The three technical replicates were checked for outliers with the Grubbs Test [29] with the GraphPad online tool (https://www.graphpad.com/quickcalcs/Grubbs1.cfm, accessed on 22 December 2020), whereas one outlier was found in the technical replicates of RPL32, which was excluded from further calculations. Subsequent data preprocessing was performed in Microsoft Excel. The arithmetic means of the CT values of the technical replicates were calculated. The arithmetic means of all technical replicates of each gene were averaged (arithmetic mean) to create a calibrator CT. After subtracting each technical replicate from the calibrator CT, the obtained ΔCT values were corrected for primer efficiencies. The relative gene expressions of the GOIs were calculated by the ΔΔCT method, using the geometric means of the efficiency-corrected ΔCT values of RPL32 and GAPDH [30]. The obtained relative gene expressions were exported to IBM SPSS Statistics 27 for statistical analyses.

### 2.12. Statistical Analysis

Values were rounded to two decimal places [31]. Locations were numbered consecutively from 1 to 3 following the cranial–caudal axis. After raw data analysis, the multiple linear regression model (MLR) was applied to account for the complex influences on the cell cultures. All ΔΔCT data were positive and greater than zero. Furthermore, the ΔΔCT values showed a right-skewed distribution for HOX A3, D1, SOX9 and COL2 and their data fitted to square root functions. Therefore, we combined reverse score transformation with squaring (Equation (Equation 1)) to meet MLR requirements and reduce outliers [31].
(1)$$\Delta \Delta CTGOI_{\mathrm{trans}}\;=\;{\left(\Delta \Delta CTGOI_{\mathrm{value}}-\Delta \Delta CTGOI_{\max\;\mathrm{value}}\right)}^{2}$$

The transformed ΔΔCT values were checked for normality with the Shapiro–Wilk test. Only HOX D1 showed a normal distribution. We performed a multiple linear regression model (MLR), in which location and cultivation times were set as predictors. The transformed ΔΔCT were sorted by cultivation time [31]. Furthermore, a one-way ANOVA was performed to check whether the transformed ΔΔCT values differed significantly between locations at T1 and T2, as this statistical test is robust to violations of the normal distribution regardless of group size [32]. Variance homogeneity was tested with Levene’s test. In the case of significance, the Bonferroni post hoc test was performed [31]. To determine whether ΔΔCT values differed significantly between T1 and T2, the differences of the paired values were checked for normal distribution with the Shapiro–Wilk test, and a paired t-test was performed for each location.

## 3. Results

### 3.1. Cell Culture

Chondrocytes of P0 were small with an either round or polygonal shape in all cultures. After the first subcultivation, the cells became increasingly spindle-shaped in all cultures, whereby nasal chondrocytes dedifferentiated faster than articular chondrocytes. After the third subcultivation, almost all cells were spindle-shaped. Additionally, enlarged and irregular shaped chondrocytes appeared (Figure A1). The cell yields are provided as Appendix A.

### 3.2. Descriptive Statistics

Mean values, standard deviations (SD) and medians (Md) of the ΔΔCT values are presented for each gene and localization in Table 2 for measurement time point T1 and in Table 3 for measurement time point T2. The distributions of ΔΔCT values around the respective median distributions are shown in box plots for each location in T1 and T2 (Figure 1).

### 3.3. Multiple Linear Regression

ΔΔCT values revealed individual gene expression levels in the HOX genes (Figure A2). Therefore, ID was included as a covariate. MLR data are presented as recommended [33] in Table 4. Regarding the HOX genes, significant effects were found for ID, whereas neither cultivation time nor location were significant. SOX9 values were not associated with the predictors or the covariate. COL2 was affected significantly by location and cultivation time, but not by the ID (Table 4). The positive coefficient of the transformed COL2 data confirms the descreasing COL2 levels over the cranio–caudal axis between the time points shown in Figure 1.

The MLR model was significant for HOX A3, HOX D1 and COL2. The adjusted R${}^{2}$ was 0.51 for HOX A3, 0.44 for D1 and 0.39 for COL2, indicating a good fit of the MLR to the data, as a large proportion of variance is accounted for by the predictors and the covariate. HOX D8 and SOX9 were not significant. The adjusted R${}^{2}$ of HOX D8 reveals that the chosen MLR model has a low goodness of fit for this gene (Table 5).

### 3.4. One-Way ANOVA

Levene’s test showed that equal variances could be assumed for all transformed ΔΔCT values of HOX genes and SOX9 in T1 and T2. Levene’s test indicates equal variances for transformed ΔΔCT values of COL2 in T1 but unequal variances for this variable in T2. The transformed ΔΔCT of COL2 violated the homogeneity of variance assumption in T2, and therefore, ANOVA was not performed for COL2 at this time point (Table 6).

One-way ANOVA revealed no significant differences in the ΔΔCT values of SOX9 and HOX A3, D1 and D8 at T1 and T2. The expression of COL2 is significant at T1 (Table 7). Bonferroni-adjusted post hoc analysis revealed a significant difference between the nose and front fetlock (*p* = 0.02, mean difference= −520.22, 95% CI [−979.59, -60.85]) as well as the nose and hind fetlock (*p* = 0.03, mean difference= −500.97, 95% CI [−960.34, −41.60]) of the transformed ΔΔCT COL2. In contrast, the transformed ΔΔCT for COL2 was not significant between the two articular chondrocytes.

### 3.5. Paired t-Test

The differences of the ΔΔCT values of T1 and T2 were normally distributed (*p* > 0.05) over all locations and for all paired values. Only COL2 differed significantly between T1 and T2 in each location (Table 8), confirming the results of the MLR model, in which only COL2 showed a significant change over time.

## 4. Discussion

Pelttari et al. (2014) observed local differences in the HOX genes C4, C5, C8 and D3 between nasal and articular chondrocytes in humans and goats. In contrast to articular chondrocytes, these genes were not or barely expressed in nasal chondrocytes in both species. Therefore, they termed the nasal HOX gene profile “HOX negative” [19].

In the present study, no significant differences were detected in the gene expression of HOX A3, D1 and D8 between nasal and articular chondrocytes. However, we do not consider our results as contradictory to the study in goats and humans [19], as HOX genes define cellular identity and tissue specific functions [34,35]. Furthermore, both articular and nasal chondrocytes form hyaline cartilage. Consequently, it is likely that there are similarities in the HOX gene profiles of chondrocytes from different locations. Our results for HOX A3, D1 and D8 gene expression using equine chondrocytes confirm this assumption.

In contrast to the studies on human and caprine chondrocytes [19], no “HOX-negative” gene profile was detected. However, this statement must be interpreted carefully, as we were not able to generate a complete HOX gene profile for the three locations, since the NCBI database provided predicted targets, only. Following the MIQE guidelines [28], only HOX A3, D1 and D8 could be used for this study. Despite the limited number of genes that were examined, we assume high validity, since HOX genes are evolutionarily conserved [36]. Nevertheless, complete information on the gene sequences and functions are needed to make reliable statements. Since equus caballus is a challenging and informative animal model in chondrocyte-related translational research [9], there is a knowledge gap that should be filled in the interest of future research addressing similar issues.

While examining the influence of cultivation time on the HOX gene profile of cultivated chondrocytes, no significant differences between the measurements T1 and T2 were found. Similar results were observed in human nasal and articular chondrocytes during expansion in a monolayer culture [19]. Given the similar HOX gene expression between the three locations and their stability during cultivation, we conclude that nasal chondrocytes up to the second passage may represent a potential target for future research on therapy options for articular cartilage. However, it must be considered that gene profiles might be different between native cartilage and cultivated chondrocytes as cell dissociation leads to molecular changes, which results in poor RNA quality [37]. In fact, due to the digestion of the cartilage, we obtained low RIN values (under 5) for several RNA samples (data not shown), which were therefore excluded from the study [38]. The reported differences in the gene expression of HOX C5, D3 and D8 between native human cartilage and the derived chondrocytes [19] may also be a consequence of the cell isolation step. In our study, the collected cartilage samples were enzymatically digested to obtain the required cell numbers for the experiments. Therefore, there was not enough tissue left to determine the HOX gene profile in native cartilage. Cells after the second passage were not used in our studies because cells of passage 3 showed signs of senescence [39] and morphological changes due to dedifferentiation [40].

In humans, nasal cartilage showed no expression of HOX D8, whereas this gene was expressed in caprine chondrocytes. Our results confirm species-specific differences, as horses express the HOX D8 gene in nasal cartilage. Moreover, in horses, HOX A3 was active in all three locations, whereas human chondrocytes were negative for this gene regardless of location [19]. Different gene expression levels between different species is known as evolutionary adaptation and has already been described for bats and dolphins [41].

Our study showed that HOX gene profiles even differ significantly within a species with respect to their individual expression level. Given there were only limited anamnestic and clinical data on the horses and only seven animals of different age and breed were available, the dataset does not allow one to draw any conclusions on the causes of this individuality. Since HOX genes are known to be influenced by age, physical activity [42], hormonal status [43,44], and even by diseases (e.g., osteoarthritic cartilage expressed higher HOX A13 levels in humans [45]), there are certainly several factors that affected each individual.

In contrast to the HOX genes, the chondrocytes marker genes SOX9 and COL2 were not affected by the individual. COL2 showed significant relations to location and cultivation time in the performed statistics. Nasal chondrocytes showed high median expression levels at T1 that decreased to COL2 levels of the articular chondrocytes in T2 (Figure 1) and shows that the equine nasal chondrocytes undergo fast dedifferentiation between T1 an T2. Furthermore, the measurement of the COL2 levels confirmed our observations of the nasal cell cultures, that already showed a high proportion of spindle-shaped cells after first subcultivation. The significant reduction in COL2 level in combination with the stable expression of SOX9 expression confirms the purity of the cultures [46]. In an in vitro study with human osteoarthritic chondrocytes, COL2 production could be re-initiated over more than seven passages after retroviral SOX9 transduction and pellet cultivation with the growth factors IGF-1 and TGF-**ß**3. In the same publication, it was noted that sufficient SOX9 production is present in early passages (under four), and thus the reactivation of the chondrogenic phenotype may be possible in chondrogenic cultures [26]. In an in vivo study in goats, AC and NCs were expanded in chondrogenic cultures before autologous implantation in articular defects. After three and six months, joint defects treated with NCs exhibited higher O’Driscoll scores in comparison with the articular defects that were treated with ACs [19]. In our study, robust SOX9 levels up to T2 were found in all locations, which might indicate that the re-differentiation of all chondrocytes is possible up to T2. Considering the higher basal COL2 expression of equine NCs that we observed and the results reported in the study of Pelttari et al. (2014), the nasal cartilage is an interesting target for further research of therapeutic options for cartilage joint defects in horses. In our study, we see no evidence that the HOX genes of the different locations correlate with chondrocyte markers.

Although SOX9 and COL2 are reliable parameters for the differentiation status of chondrocytes in vitro, other factors such as Collagen I or proteoglycans [46] should be analyzed to assess the differentiation status of chondrocytes more specifically. Moreover, a gene expression profile does not reflect the proteins that are contained in the extracellular matrix. In former studies, the analyses of extracellular matrix proteins [17] or histology [46] were performed to determine the differentiation status of chondrocyte cultures.

Regarding cultivation, recent studies show that conditions could be optimized in favor of the functional preservation of chondrocytes. In recent years, growth factors were used predominantly to maintain the chondrocyte’s gene expression patterns. Growth factors affect many regulatory genes [19,47,48], including HOX genes. Without a reliable knowledge of signaling pathways, the treatment of cell cultures with growth factors always bears the risk of unwanted clinical outcomes, especially given that HOX genes might have a functional role in cancer development [35]. A solution for this might be the use of physical forces on cultivated chondrocytes which have been studied in several publications already. It was found that the production of COL2 and SOX9 could be maintained at higher passages if chondrocyte cultures were exposed to moderate pressure [49,50]. New findings even suggest that the components of the medium and the type of culture [46] might have an influence on the re-differentiation capacity of chondrocytes. Another study revealed a re-differentiation effect in chondrocytes cultured on inactivated collagen matrix without growth factor supplementation [51].

## 5. Conclusions

In this study, we demonstrated that cultured nasal chondrocytes may be of interest for research on potential therapeutic options for equine joint defects. The HOX genes’ expression of A3, D1 and D8 between articular and nasal chondrocytes are similar and stable during cultivation. Furthermore, equine nasal chondrocytes showed a higher basal COL2 expression than articular chondrocytes. This study also confirms that chondrocytes undergo dedifferentiation in a monolayer culture, and that the expression levels of the three investigated HOX genes are more dependent on the individual horse, e.g., in terms of age, than on location or cultivation time.

## Figures and Tables

**Figure 1 animals-11-02542-f001:**
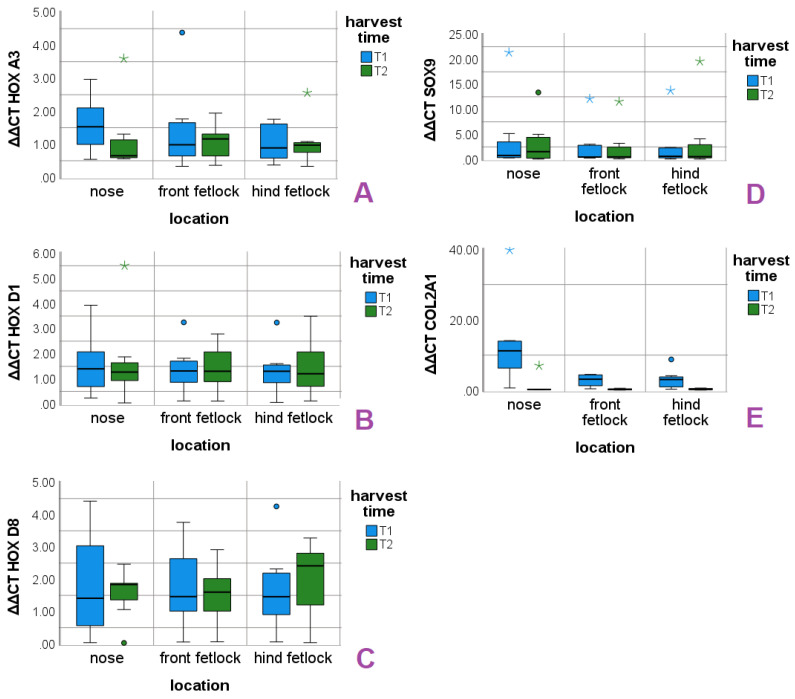
Expression of the HOX genes A3 (**A**), D1 (**B**), D8 (**C**) and the chondrocyte marker genes SOX9 (**D**) and COL2 (**E**) in the three locations at T1 and T2. Dots and asterisks represent outliers and extreme outliers, respectively.

**Table 1 animals-11-02542-t001:** List of used primers.

	NCBI		Product	Tm
Gene	Acc. No.	Primer Sequence (5${}^{\prime }$-3${}^{\prime }$)	Size bp	°C
HOX genes				
HOX A3	XM_0147	F: TCTGAAAAATGGCTCCCTCC	118	58
	39059.2	R: CGCAGACCTCGAAAGATGT		57
HOX D1	XM_0236	F: CCCCCAAGAAAAGCAAACTC	81	58
	22046.1	R: GTTGCTTGGTGCTGAAGTTA		56
HOX D8	XM_0236	F: GATGAGACCACAAGCTCCT	119	57
	22042.1	R: CGATCCTTCTCTTCCTGGTC		60
chondrocyte marker genes				
SOX9	XM_0236	F: CTCTGGAGACTGCTGAACG	129	59
	52130.1	R: GTTCTTCACCGACTTCCTCC		60
COL2	NM_0010	F: CTTCCACTTCAGCTATGGAGATG	117	63
	81764.1	R: TTCTTGCAGTGGTAGGTGATG		59
reference genes				
GAPDH	NM_0011	F: AGCTCATTTCCTGGTATGACA	120	57
	63856.1	R: CCTTCTCTTGCTGGGTGATT		58
RPL32	XM_00149	F: TGTGCAACAAATCGTACTGT	112	54
	2042.6	R: GCATTGGGATTGGTGATTCT		56

**Table 2 animals-11-02542-t002:** Descriptive statistics for T1.

T1									
	**Nose**			**Front Fetlock**			**Hind Fetlock**		
**ΔΔCT**	**Mean**	**SD**	**Md**	**Mean**	**SD**	**Md**	**Mean**	**SD**	**Md**
HOX A3	1.61	0.89	1.53	1.47	1.39	0.98	1.06	0.60	0.89
HOX D1	1.59	1.27	1.40	1.40	1.00	1.32	1.34	1.00	1.30
HOX D8	1.87	1.78	1.41	1.80	1.35	1.46	1.71	1.39	1.46
SOX9	4.31	7.72	0.79	2.76	4.28	0.48	2.81	4.92	0.62
COL2	13.18	12.88	11.22	2.80	1.75	3.18	3.13	2.89	3.09

**Table 3 animals-11-02542-t003:** Descriptive statistics for T2.

T2									
	**Nose**			**Front Fetlock**			**Hind Fetlock**		
**ΔΔCT**	**Mean**	**SD**	**Md**	**Mean**	**SD**	**Md**	**Mean**	**SD**	**Md**
HOX A3	1.18	1.10	0.65	1.05	0.56	1.16	1.07	0.71	0.97
HOX D1	1.71	1.77	1.24	1.45	0.92	1.30	1.48	1.15	1.20
HOX D8	1.54	0.79	1.84	1.52	0.97	1.60	1.96	1.24	2.41
SOX9	3.47	4.77	1.55	2.53	4.13	0.48	3.82	7.10	0.54
COL2	1.24	2.54	0.29	0.32	0.20	0.24	0.36	0.23	0.33

**Table 4 animals-11-02542-t004:** MLR coefficients (B) and corrected coefficients (**ß**) of constant, predictors, and covariates are presented as well as their significance levels (*p*) for the transformed ΔΔCT values of each GOI.

GOI ΔΔCT_trans_		B [95% CI]	ß	*p*
HOX A3	constant	11.82 [6.81, 16.83]		0.00
	location	0.78 [−0.74, 2.29]	0.15	0.31
	cultivation time	0.07 [−0.07, 0.208]	0.14	0.34
	ID	−1.00 [−1.62, −0.37]	−0.46	0.00
HOX D1	constant	26.69 [19.06, 34.33]		0.00
	location	0.42 [−1.89, 2.74]	0.05	0.71
	cultivation time	−0.00 [−0.22, 0.21]	−0.00	0.98
	ID	−2.53 [−3.48, −1.58]	−0.66	0.00
HOX D8	constant	6.21 [−1.40, 13.83]		0.11
	location	−0.42 [−2.73, 1.89]	−0.06	0.71
	cultivation time	−0.04 [−0.25, 0.175]	−0.06	0.72
	ID	1.01 [0.06, 1.95]	0.33	0.04
SOX9	constant	364.16 [185.62, 542.69]		0.00
	location	8.59 [−45.49, 62.68]	0.05	0.75
	cultivation time	−0.53 [−5.55, 4.48]	−0.04	0.83
	ID	−4.15 [−26.31, 18.01]	−0.06	0.71
COL2	constant	632.11 [288.89, 975.33]		0.01
	location	140.86 [36.88, 244.84]	0.35	0.01
	cultivation time	19.17 [9.53, 28.81]	0.51	0.00
	ID	21.93 [−20.67, 64.53]	0.13	0.30

**Table 5 animals-11-02542-t005:** F-values for MLR with degrees of freedom (df), significance levels (*p*) and quality of the MLR model for each GOI without (R${}^{2}$) and with correction (adjusted R${}^{2}$) for the parameters location, cultivation time and ID.

GOI ΔΔCT_trans_	F (df = 3; 38)	*p*	R${}^{2}$	adj R${}^{2}$
HOX A3	4.38	0.01	0.51	0.26
HOX D1	9.83	0.00	0.66	0.44
HOX D8	1.70	0.18	0.34	0.12
SOX9	0.09	0.96	0.09	0.01
COL2	8.07	0.00	0.62	0.39

**Table 6 animals-11-02542-t006:** Levene’s test with F-values and significance levels (*p*) for the time points T1 and T2.

	T1		T2	
**GOI ΔΔCT_trans_**	**F (df = 2; 18)**	**p**	**F (df = 2; 18)**	* **p** *
HOX A3	0.35	0.71	0.26	0.76
HOX D1	0.20	0.82	0.1	0.91
HOX D8	0.21	0.82	0.64	0.54
SOX9	0.24	0.79	0.15	0.86
COL2	2.85	0.08	4.44	0.03

**Table 7 animals-11-02542-t007:** One-way ANOVA with F-values and significance levels (*p*) for the time points T1 and T2 between the three locations.

	T1		T2	
**GOI ΔΔCT_trans_**	**F (df = 2; 18)**	**p**	**F (df = 2; 18)**	* **p** *
HOX A3	0.68	0.52	0.00	0.99
HOX D1	0.06	0.94	0.00	0.99
HOX D8	0.02	0.98	0.18	0.83
SOX9	0.05	0.95	0.08	0.93
COL2	5.74	0.01		

**Table 8 animals-11-02542-t008:** Results of two-tailed paired t-test for ΔΔCT_trans_ in each location, with confidential interval (CI), difference (T), degrees of freedom (df) and significance levels (*p*).

GOI ΔΔCT_trans_	95% CI			
T1–T2 Paired	Lower; Upper	T	df	*p* (Two-Tailed)
nose = location 1				
HOX A3	−7.18; 1.37	−1.66	6	0.15
HOX D1	−4.02; 3.33	−0.23	6	0.83
HOX D8	−4.20; 5.04	0.224	6	0.83
SOX9	−40.95; 45.39	0.13	6	0.90
COL2	−1135.97; −147.22	−3.18	6	0.02
front fetlock = location 2				
HOX A3	−3.69; 1.23	−1.23	6	0.27
HOX D1	−3.83; 4.81	0.28	6	0.79
HOX D8	−4.01; 2.49	−0.57	6	0.59
SOX9	−20.51; 4.96	−1.49	6	0.19
COL2	−305.71; −69.57	−3.89	6	0.01
hind fetlock = location 3				
HOX A3	−3.02; 2.87	−0.06	6	0.95
HOX D1	−2.95; 4.68	0.56	6	0.60
HOX D8	−0.74; 4.03	1.69	6	0.14
SOX9	−18.36; 46.35	1.06	6	0.33
COL2	−386.53; −19.65	−2.71	6	0.04

## Data Availability

Data are deposited in the Institute of Biochemistry and are available on request to the corresponding author.

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
