# Peer review of "HOX Gene Expressions in Cultured Articular and Nasal Equine Chondrocytes"

_animals, 2021, doi:10.3390/ani11092542_

Round 1
Reviewer 1 Report
Small regenerative capacity of articular cartilage requires the application of in vitro expanded chondrocytes or a variety of cell types of mesenchymal origin which have chondrogenic potential in regenerative treatment of damaged joint areas. Every information about any new cell source needful for transplantation purposes is always very valuable. Therefore I found the content of the manuscript very practical and interesting. The methods used are well described. Especially of note is the careful settlement of RT-PCR conditions for HOX genes and COL-2 and SOX9 markers allowing to obtain reliable molecular data at the initial stages of chondrocyte expansion in 2-D culture.
My single comment refers to the Conclusions chapter (lines 317 to 320)
“ the expression levels of the three investigated HOX genes are more dependent on the individual horse than on location or cultivation time.”
I think that it will be good to mention that the age is also the strong factor of variability of expression patterns in case of investigated genes.
In the method chapter (line 68 and 69) the described age differences of the donors of cells are very large.
Reviewer 2 Report
This manuscript aims at evaluating the potential of chondrocytes derived from the nasal septum for autologous chondrocyte implantation (ACI) in horses by comparing their gene expression profile whit that of articular chondrocytes. This is an innovative approach, as nasal septum would provide advantages over articular cartilage for tissue harvesting, and is well based on previous studies in humans and other species, providing data for the horse for the first time. The manuscript is overall well written and the study design is simple but appropriate for the goal of the study. Whereas other methodologies could have been used to complement the results, the authors acknowledge that this is a preliminary study that warrants further research. Methodology regarding chondrocyte obtainment and culture, as well as qPCR, is very detailed. However, information regarding data analysis and statistics mix both methods and results and need further organisation. The results section would benefit from including additional information. Discussion is well organised and clear and, as well as conclusions, remarks main findings and put them into context with clear take-home messages. Please find below more specific comments regarding some parts of the manuscript that need revision before further considering it for publication:
Material and methods
2.5. Cell passage
- Line 116: do authors mean trypsin-EDTA? Please revise.
2.11. Data Analysis
- Lines 189-192: Please clarify how ddCt values for T1 were calculated as presented in table 2 and figure 1. If T1 was used as calibrator, the ddCt values for T1 should be 0 in all cases: ddCt = dCt sample - dCt calibrator; if sample and calibrator are the same, the calibrator does not vary over itself. In addition, provided that the main goal of the study is to compare nasal and articular chondrocytes to explore the potential of the first ones for ACI therapy, it could be more appropriate to use fetlock chondrocytes as calibrator.
- Tables 2 and 3, and figure 1, and the information in the main text regarding these items, pertain to results rather than to methods. Please modify accordingly. Methods should only contain how analysis (qPCR data and statistics) were done, but not the outcomes.
- Figure 1: please explain symbols (asterisks and dots) used in the figure caption, e.g. outliers?
Results
- In addition to the gene expression study, I would recommend including information on the phenotype of chondrocytes, i.e. morphology in culture and changes noted along time and among conditions (with accompanying photographs). Data about the yield obtained and viability at each passage would also complement the results. For example, in the discussion (lines 262-264): 'Cells after the second passage were not used in our studies because cells of passage 4 showed morphological changes due to de-differentiation': it is needed to include such data under results to discuss it.
- Statistical analysis (concerning how it was performed) should be presented under the Methods section, in a separate subheading 'Statistical Analysis'.
- Please provide reference to support the transformation strategy followed for ddCt data.
- Statistics performed are focused on analysing the effect of time, location and individual. Whereas this is an interesting approach, I strongly recommend completing statistical analysis by showing whether there are or not significant differences among groups for each gene, i.e among locations at each time-point and between time-points within each location. Descriptive statistics show the data for each condition and one can see, for example, that COL2A1 gene expression is clearly higher at T1 than at T2 for all the conditions (especially in figure 1, which is a more graphic representation). Therefore, whereas it is interesting to know that location and time influence the expression of COL2A1, completing these results with the direction in which these changes take place will strengthen and clarify the results (please note that COL2A1 has been used as example, but this comment applies to all genes). For example, in the discussion it is stated:
- Lines 218-219 'In the present study, no significant differences were detected in gene expression of HOX A3, D1 and D8 between nasal and articular chondrocytes'
- Lines 238-239: ‘while examining the influence of cultivation time on the HOX gene profile of cultivated chondrocytes, no significant differences between the measurements T1 and T2 were found’.
These sentences imply that comparisons among locations and between time-points have been performed, but results are not presented in this way. Please complete the results by providing both appropriate figures and informative description in the text, using sentences similar to those pointed out in the discussion.
- Lines 210-212 and table 5: Please further develop these lines to clarify what imply the results of this statistics shown in table 5.
Discussion
- Lines 267-268: were joints from which cartilage was harvested examined for the presence of osteoarthritis or other joint disorder, at least visually? If so, this information should be provided (as joints free of abnormalities, or stating in how many horses were abnormalities found and of which type and severity).
- Lines 292-297: Please provide explanation for why these analyses were not performed in the current study (lack of enough sample, budget limitations...).
